# Whole genome DNA methylation patterns in tissue and cfDNA associated with fibrosis reflect the complex signature of MASLD

Jongseong Ahn[1☯], Soyeon Kim[2,3☯], Jae Yoon Jeong[4☯], Sunghoon Heo[1], Kyung-hee Pyo[2,3], Eun-Ae Shin[2,3], Wonsik Kim[2,3], Jae-Ho Lee[2,3], Na Ryung Choi[4], Han Ah Lee[5], Hwang-Phill Kim[1], Sang-Hyun Song[1], Hwi Young Kim[4¤*], Tae-You Kim[1,6*], Jung Weon Lee [2,3*]

1 IMBdx, Seoul, Republic of Korea, 2 Department of Pharmacy, College of Pharmacy, Seoul National University, Seoul, Republic of Korea, 3 Research Institute of Pharmaceutical Sciences, College of Pharmacy, Seoul National University, Seoul, Republic of Korea, 4 Department of Internal Medicine, Ewha Womans University College of Medicine, Division of Gastroenterology and Hepatology, Ewha Womans University Mokdong Hospital, Seoul, Republic of Korea, 5 Department of Internal Medicine, Chung-Ang University College of Medicine, Seoul, Republic of Korea, 6 Department of Internal Medicine, Seoul National University Hospital, Seoul, Republic of Korea

☯ These authors contributed equally to this work.
¤ Current address: Division of Gastroenterology and Hepatology, University of Nebraska Medical Center, Omaha, Nebraska, United States of America
* hwiyoung@gmail.com (HYK), kimty@snu.ac.kr (T-YK), jwl@snu.ac.kr (JWL)

## Abstract

Metabolic dysfunction-associated steatotic liver disease (MASLD) can progress to steatohepatitis, being associated with inflammation, fibrosis, and immune cell interactions. Recent studies have reported an association between DNA methylation (DNAm) and MASLD. However, the relationship between DNAm and MASLD-related fibrosis is limited to liver tissue alone or focused on particular CpG sites. Moreover, despite the widely recognized sex differences in MASLD, studies that account for this variable remain limited. We performed whole-genome methylation sequencing (WGMS) on liver tissue and cell-free DNA (cfDNA) from patients with MASLD to investigate the association between fibrosis and DNAm. After initial filtering, the tissue and cfDNA data were further grouped by sex, and DNAm sites exceeding the minimum threshold for association with fibrosis were selected. Pathway analysis based on the genomic locations of CpG bins revealed both overlapping and distinct MASLD- and fibrosis-associated pathways across tissue and cfDNA, between males and females, and between intragenic and intergenic regions. Hepatic tissue deconvolution indicated the presence of immune cells and confirmed an increase in liver-derived cfDNA associated with fibrosis in cfDNA samples. Our study demonstrates the potential of WGMS as a platform for comprehensively observing the complex patterns of MASLD alterations.

**Data availability statement:** Raw data for this study were generated at IMBdx, Inc, using the clinical samples proved by IRB (Ewha Womans University Mokdong Hospital Institutional Review Board, irb-mok@eumc.ac.kr). Upon reasonable requests written to the data access at R&D Center, IMBdx, Inc (https://www.imbdx.com/other/customer.php) or to Ewha Womans University Mokdong Hospital Institutional Review Board, the data and sample material information can be available.

**Funding:** This work was supported by the National Research Foundation of Korea (NRF) funded by Basic Science Research Program through the National Research Foundation of Korea (NRF) funded by the Ministry of Science, ICT & Future Planning (NRF-2018R1A5A2025286 to HYK, NRF-2017M3A9A7050610 to T-YK, and NRF-2020R1A2C3008993 and NRF-2021M3A9D3024752 to JWL) and by a grant of Korean ARPA-H project through the Koreas Health Industry Development Institute (KHIDI), funded by the Misnistry of Health & welfare, Republic of Korea (RS-2024-00512207 to T-YK). The funders had no role in study design, data collection and analysis, decision to publish, or preparation of the manuscript. These funding details include all funding or source of supports (external or internal of our authors' organizations) received during this study. There was no additional external funding received for this study.

**Competing interests:** I have read the journal's policy and the authors of this manuscript have the following competing interests: JA, SH, S-HS, and H-PK are employee of IMBdx, Inc. T-YK is co-founder of IMBdx. The other authors declare no potential conflicts of interest. this manuscript has not been published and is not under consideration for publication elsewhere.

## Introduction

DNA methylation (DNAm) primarily occurs as methylation at cytosine residues preceding guanine in humans; these positions are commonly referred to as CpG sites. Although these CpG sites are depleted compared with their natural occurrence across the genome, they still tend to cluster in specific regions [1]. Regions with a high density of CpG sites are known as CpG islands and have been the focus of significant interest due to their association with gene expression [2]. Hypermethylation, characterized by an increased proportion of methylated CpG sites in these regions, is thought to inhibit gene expression by blocking the access of RNA polymerase or transcription factors. Based on this traditional perspective, methods focusing on CpG density such as reduced representation bisulfite sequencing (RRBS) have been developed and proven to be useful [3].

This CpG island model has recently encountered limitations, as various methylation patterns that cannot be explained by the existing model have been increasingly discovered. Notably, changes in regions adjacent to CpG islands have been found to vary more specifically based on cancer presence [4]. To address these distance-related changes, terms such as shore (approximately 2 kb), shelf (approximately 4 kb), and open sea (> 4 kb) have been introduced. Additionally, the understanding of DNA methylation has expanded from focusing solely on regions near gene transcription start sites (TSSs) to include gene body methylation [5]. The expansion of the observation window has necessitated the use of 450K and EPIC (850K) methylation arrays [6], leading to the detection of a broader range of changes and ultimately culminating in the need for whole-genome methylation sequencing (WGMS) to capture comprehensive genomic methylation changes for applications in environmental studies [7], exercise [8], and various diseases [9]. Notably, the observation that most cell type–specific methylation markers are located in regions not covered by existing arrays underscores the necessity of utilizing WGMS for a more comprehensive analysis [10].

Metabolic dysfunction-associated steatotic liver disease (MASLD), previously termed non-alcoholic fatty liver disease (NAFLD), encompasses the accumulation of liver fat due to non-alcoholic causes and various conditions, including metabolic dysfunctions [11]. Its prevalence is steadily increasing globally, and it is known to be associated with obesity [12], type 2 diabetes (T2D) [13], and cardiovascular diseases [14]. Severe forms of MASLD, such as metabolic dysfunction-associated steatohepatitis (MASH, previously termed non-alcoholic steatohepatitis NASH), can progress to or associate with fibrosis and ultimately lead to cirrhosis or liver cancer, which has a lower survival rate [15]. Thus, there is a clear clinical need for the effective detection of MASLD changes. However, traditional liver biopsy [16], the gold standard for diagnosis, is invasive and carries significant drawbacks, such as inconvenience and potential side effects, leading to reluctance associated with its use and perpetuating a cycle of limited understanding of MASLD. In response to these limitations, various non-invasive methods for MASLD assessment have been researched [17]. Whereas previous methods primarily relied on protein-based approaches, recent developments have highlighted

the potential of using cell-free DNA (cfDNA), DNA fragments that circulate freely in the blood. Although there are various interpretations regarding the source of cfDNA [18], the correlation between the origin of cfDNA detected and organs damaged due to disease supports the hypothesis that cfDNA is residual material from cells that have died for various reasons [19]. Consequently, normal individuals have low levels of cfDNA, primarily derived from blood cells, but in cases of cancer or inflammation, cfDNA originating from affected tissues can be detected, making it a promising target for non-invasive monitoring [20]. Analysis of cfDNA through DNAm patterns can reveal tissue-specific DNAm profiles, making DNA methylation pattern analysis a preferred method. The understanding that MASLD is a complex disease involving not only liver cells but also various immune cells highlights the necessity of cell-type deconvolution through methylation analysis [21].

In this study, we investigated whole-genome methylation patterns in both tissue and blood samples to identify their associations with MASLD features. We first observed their associations with fibrotic features that correlated with MASLD development. The associations, stratified by sex and sample source, revealed intriguing patterns consistent with previously reported differences related to sex. Furthermore, the broader observational scope provided by WGMS extends beyond the limited focus on CpG islands, uncovering direct and indirect pathway associations across a wider range of CpG sites. Our study demonstrates that WGMS can be a suitable approach for tracking the complex patterns of MASLD and highlights the potential for non-invasive MASLD diagnosis using blood-based methods.

## Materials and methods

### Study population and sample collection

Tissue or blood samples were collected from patients with MASLD, LC and HCC at Ewha Womans University Mokdong Hospital following IRB approval (EUMC 2016-07-052). Due to the lack of main clinical information, LC and HCC samples were used solely as reference data for comparison in the cell type deconvolution analysis.

### Sample preparation and WGMS

All WGMS and data preprocessing were conducted following the same procedures described in previous studies [22]. Briefly, tissue DNA and cfDNA were extracted from tissue and plasma samples using dedicated kits. The DNA samples, after concentration measurement, were processed using the IMBdx AlphaLiquid® Screening platform protocol, where cytosine was converted depending on the presence of methylation. Raw sequencing data (~ 100 GB) sequentially underwent trimming, alignment, sorting, indexing, deduplication, and methylation report generation at individual CpG sites. The resulting deduplicated BAM files and reports were used for analysis.

### Selection of significant CpG sites through correlation with fibrosis and cross-matching between groups

Based on previously reported associations related to inter-CpG distance [23], all CpG sites were binned together if the distance between adjacent sites was less than 65 base pairs. The beta value for each bin was defined as the ratio of methylated CpGs to the total number of observed CpGs within the group like average methylation fraction [24]. The remaining samples, excluding non-MASLD cases, were divided into four groups based on sex and sample type. Kendall correlation analysis between the beta values of each sample and fibrosis stage was performed for all individual CpG group using the kendalltau function from Python (ver 3.10.6)'s SciPy (ver 1.10.0) module, which calculated both the correlation coefficients and p-values. Bins were excluded if they were observed in fewer than four individuals or if the values converged to fewer than three unique values. At each p-value threshold, values less than 0.05 were considered statistically significant. For the retained CpG sites, only those simultaneously found in the same sex or sample type were considered valid. Each intersection was named according to the shared sex or sample type. Intersections shared by all groups were named "All". Unless otherwise specified, all subsequent analyses were conducted in an R environment (r-base; ver 4.2.1) on JupyterLab (ver 3.5.0) running on conda (ver 24.3.0).

## Pathway analysis for intersection by sex and sample type

All subsequent pathway analyses were conducted by inputting the extracted gene lists into the web-based tool Enrichr [25] (https://maayanlab.cloud/Enrichr), with settings fixed to the default option, which includes the top 100 most relevant genes. For tissue samples, bins were selected based on proximity to CpG islands or a distance of less than 1500 base pairs from TSS. For the other groups, the same cut-off criteria were applied; however, to ensure sufficient input gene counts, bins were selected based on an OR condition for the defined thresholds. In the extended analysis, bins meeting the threshold of Kendall p-value < 0.01 and showing chromHMM status [26] differences between GM12878 and HepG2 were selected, and the corresponding gene list was used as input for downstream analyses. The results were based on the Kyoto Encyclopedia of Genes and Genomes (KEGG) 2021 Human database in Enrichr. KEGG served as the primary database for pathway analysis. When necessary for cross-validation, additional databases were incorporated, including WikiPathways 2024 Human and MSigDB Hallmark 2020.

## Intergenic CpG site analysis

RefSeq gene data (hg19.ncbiRefSeq.gtf.gz) were downloaded from the UCSC Genome Browser (https://hgdownload.soe.ucsc.edu/goldenPath/hg19). After converting the data to BED format, the valid CpG bins identified earlier were annotated with their nearest gene using the "closest" function from the bedtoolsr package (ver 2.30.0.2). For CpG sites identified as intergenic, chromHMM status comparison was performed between GM12878 and HepG2 cell line data using the Core 15-state model from ENCODE/Broad, which was downloaded from the UCSC genome browser database, the same source used for Refseq data. The chromHMM status of each CpG bin was determined using the "intersect" function in bedtoolsr. The proportion of each chromatin state within each cell type was calculated and compared between fibrosis related groups or not. A gene list composed of genes nearest to bins showing >1.5-fold enrichment in chromatin state changes within fibrosis-related groups was used for pathway analysis.

## Cell-type deconvolution

We used the wgbstools program (ver 0.2.0) from a previous study [10] to generate PAT files for pre-identified cell type marker regions from the BAM files of our entire sample set. Deconvolution was then performed using UXM software (ver 0.1.0) with a pre-calculated deconvolution table provided by the same study. For controls, we downloaded and used data from five hepatocyte tissue samples (excluding one with unclear sex) from the same study's data (GSE186458; Liver-Hepatocytes). Additionally, for comparison, 30 samples from healthy cfDNA data, processed through the same WGMS procedure in the previous study [22], were randomly selected, and deconvolution was performed on their BAM files following the same process. Hierarchical clustering was conducted using the ComplexHeatmap (ver 2.14.0) library in R. The associations between key clinical variables and changes in cell type–specific proportions were analyzed using multi-variable regression, implemented with the glm package in R. The significance of distribution differences between fibrosis groups for specific cell types was assessed using the stat_compare_means function from the ggpubr package (ver 0.6.0).

## Comparison of observed methylation trends with previously known transmembrane 4 L six family member 5 (TM4SF5)–related changes and databases

To contrast our findings with previously known patterns of change in MASLD, we examined CpG methylation patterns in *TM4SF5*, a gene that has been extensively studied for its functional roles. Among the CpG bins located within the gene, we specifically filtered for those located in the first exon or first intron, as methylation in these regions is known to strongly correlate with gene expression. The selected CpG bins were manually verified. For comparison, 450K array and RRBS data near the relevant CpG sites were downloaded from the same UCSC hg19 database, focusing on data from HepG2 cells, GM12878 cells, and hepatocytes. We examined the methylation data of CpG sites within the CpG islands where

the CpG sites identified in our data were located. The data values were converted to percentages according to standard definitions for consistency in comparison. Classification using all-group bins was performed with the tidymodels package in R, with xgboost selected as the modeling engine.

## Statistical analysis

The statistical analysis was performed using R (version 4.1.1) software, and graphical visualization was performed using the "ggplot2" package. Unless otherwise specified, differences between two continuous variable groups were assessed using the Wilcoxon rank-sum test, and correlations were evaluated using Kendall's tau correlation. A significance threshold of $p < 0.05$ was applied by default.

## Results

### Patient and sample characteristics

The composition of the study participants is shown in Table 1. Of the 38 individuals, 30 had both tissue and cfDNA samples available (paired), while the remaining 8 had only one of the two sample types. All MASLD samples were included in the analysis of the correlation between fibrosis stage and DNA methylation. Categorical variables for each individual are presented in Fig 1, while continuous variables are provided alongside categorical annotations in a separate table (S1 Table).

### The correlation between DNA methylation in tissue and fibrosis reflects reported fibrosis-associated patterns

We stratified the data by sex and sample type to identify CpG bins that showed correlations with fibrosis stage in two or more groups (Fig 2). To validate this grouping approach, we examined bins assigned to the "Tissue" group that were located within CpG islands at transcription start sites (TSS), which are traditionally considered closely associated with gene expression. A total of 87 bins were identified as meeting these criteria (S2 Table). Four bins showed hypomethylation patterns—typically indicative of increased expression—in both males and females. These bins were located at the TSS regions of *KIF3A* [27], *RAD21* [28], *PFKFB3* [29], and *UBB* [30], all of which have previously been reported to have direct or indirect associations with fibrosis. Pathway analysis of all 87 bins revealed statistically significant enrichment for the "Neuroactive ligand-receptor interaction" and "cGMP-PKG signaling pathway", along with relatively strong significance for the "cAMP signaling pathway" and "Regulation of lipolysis in adipocytes" (Table 2).

We further sought to characterize the overall patterns observed in the Tissue group. To exclude regions altered independently of a direct regulatory role, we conducted pathway analysis on bins from the Tissue group selected based on more stringent p-value thresholds and known chromatin alterations, rather than their positional relationship to TSS or CpG islands. This analysis revealed enrichment of pathways more directly related to fibrosis, such as the "Hippo signaling pathway" [31] and "Rap1 signaling pathway" [32]. However, the adjusted q-values did not reach statistical significance (S3 Table).

### Fibrosis-associated methylation patterns exhibit sex-specific differences

Next, we examined the methylation patterns in the "Female" and "Male" groups, each restricted to a single sex, where correlations between fibrosis and methylation were observed in both tissue and cfDNA. Similar to the analysis performed for the "Tissue" group, pathway analysis was conducted using genes associated with fibrosis-related bins located CpG islands or near TSSs (S4 Table). The "Male" group showed enrichment of pathways directly related to fibrosis, including the Notch signaling pathway [33] and Wnt signaling pathway [34]. In contrast, the "Female" group exhibited enrichment in multiple hormone- and neurotransmitter-related pathways, with a particularly strong association observed with the parathyroid hormone pathway. Additionally, we investigated the detailed methylation patterns of the Peroxisome proliferator-activated receptor gamma (*PPARG*) gene, which has been reported to be closely associated with sex-specific differences

**Table 1. Composition of Study Samples.**

| | | MASLD (n = 38) | |
| --- | --- | --- | --- |
| | | Male (n = 18) | Female (n = 20) |
| **Sample type** | Tissue-only | 2 | 4 |
| | cfDNA-only | 2 | 0 |
| | paired | 14 | 16 |
| **Fibrosis (stage)** | 0 | 0 | 2 |
| | 1 | 7 | 8 |
| | 2 | 7 | 4 |
| | 3 | 3 | 5 |
| | 4 | 1 | 1 |
| **Lobular inflammation (grade)** | 1 | 10 | 11 |
| | 2 | 6 | 9 |
| | 3 | 2 | 0 |
| **Portal_inflammation (grade)** | ND | 0 | 1 |
| | 0 | 4 | 6 |
| | 1 | 11 | 9 |
| | 2 | 3 | 4 |
| **Steatosis (grade)** | 1 | 6 | 14 |
| | 2 | 9 | 6 |
| | 3 | 3 | 0 |
| **ballooning (grade)** | 0 | 1 | 3 |
| | 1 | 12 | 11 |
| | 2 | 5 | 6 |
| **Diabetes** | No | 7 | 13 |
| | Yes | 11 | 7 |
| **Dyslipidemia** | No | 13 | 14 |
| | Yes | 5 | 6 |
| **Hypertension** | No | 14 | 11 |
| | Yes | 4 | 9 |
| **smoking** | No | 17 | 19 |
| | Yes | 1 | 1 |
| **Age** | median (min-max) | 44 (18-61) | 57.5 (21-84) |
| **BMI (kg/m²)** | | 29.5 (26.1-42.9) | 29.3 (23.5-38.7) |
| **Glucose (mg/dL)** | | 105.5 (90-163) | 104.5 (81-250) |
| **HDL (mg/dL)** | | 42.5 (26-446) | 47 (17-83) |
| **Hba1c (%)** | | **6.3 (4.4-8.6)** | **5.8 (5-8.7)** |
| **Triglyceride (mg/dL)** | | **136.5 (81-1416)** | **142.5 (85-264)** |

[35] and also linked to fibrosis [36]. For bins within *PPARG* that showed a significant correlation with fibrosis, we compared methylation distributions between high- and low-fibrosis groups within each category. Among the four bins assigned to the "Tissue" group, two showed statistically significant differences in tissue samples from both males and females (Fig 3A). In contrast, none of the bins demonstrated significant differences in cfDNA. Notably, the direction of methylation changes exhibited an overall opposite trend between males and females. In the "Male" group, one of the two bins (bin index: 1840360) showed statistically significant differences in both tissue and cfDNA, with a consistent pattern of hyper-methylation observed with increasing fibrosis severity (Fig 3B). In the "Female" group, one of the three bins (bin index:

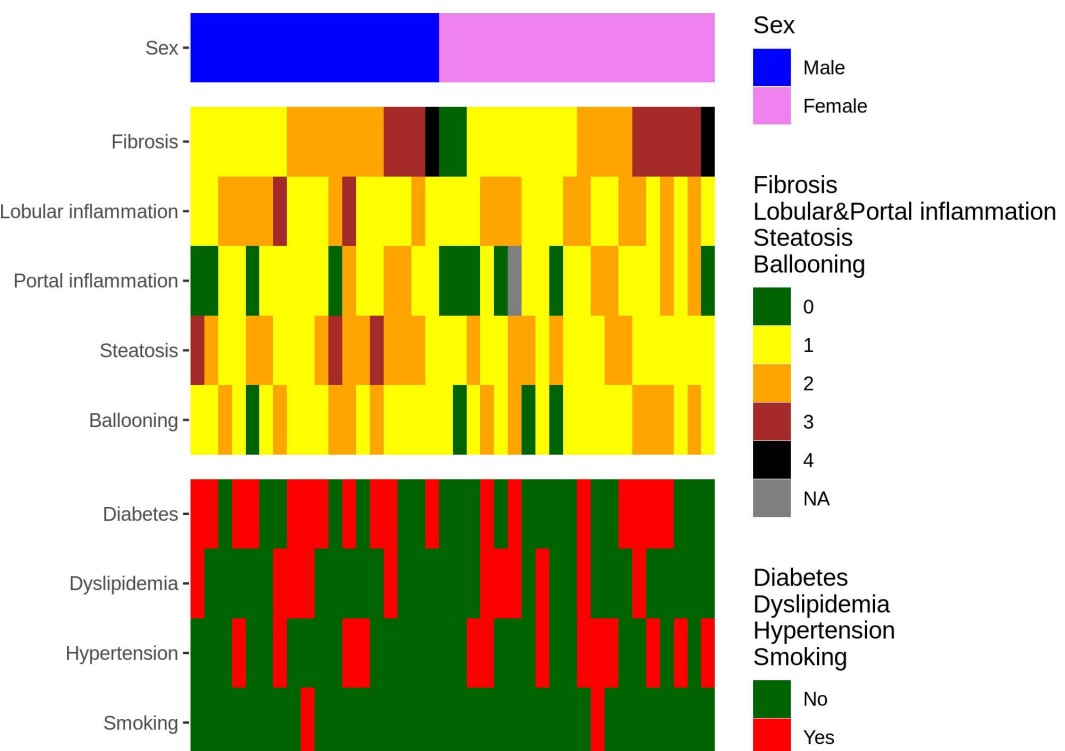

**Fig 1. Clinical characteristics of each patient.** Variables are categorized into sex, liver status-related variables, and other MASLD-associated clinical parameters. Continuous variables are provided in a separate table (S1 Table). Missing values are indicated as NA.

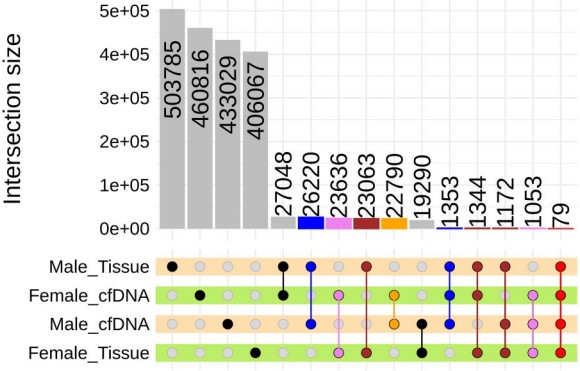

**Fig 2. Selection of valid intersecting CpG bins through cross-validation in groups between sex and tissue or cfDNA sample source.** In each of the four groups, divided by sex and sample source, CpG bins showing correlations between fibrosis and methylation were identified. Only the inter-secting groups that shared the same sex or sample source were considered valid (e.g., CpG bins observed in both male cfDNA and male tissue were classified under the "Male" group). If a sample could be assigned to three groups, tissue type was prioritized for grouping, followed by sex. Each group was represented using a distinct color: tissue (brown), male (blue), female (purple), cfDNA (orange), and all groups (red). CpG bins observed correlation across all groups were labeled as "All". Invalid groups are shaded black.

**Table 2. Table of top 10 significant p-values and q-values for pathway analysis.**

| Term | p-value | q-value |
|---|---|---|
| **Neuroactive ligand-receptor interaction** | <0.001 | 0.041 |
| **cGMP-PKG signaling pathway** | <0.001 | 0.041 |
| Regulation of lipolysis in adipocytes | 0.002 | 0.064 |
| cAMP signaling pathway | 0.002 | 0.064 |
| Glutamatergic synapse | 0.012 | 0.276 |
| Growth hormone synthesis, secretion and action | 0.013 | 0.276 |
| Calcium signaling pathway | 0.018 | 0.315 |
| Long-term potentiation | 0.032 | 0.46 |
| Renin secretion | 0.033 | 0.46 |
| Kaposi sarcoma-associated herpesvirus infection | 0.046 | 0.538 |

q-value: Benjamini-Hochberg method.

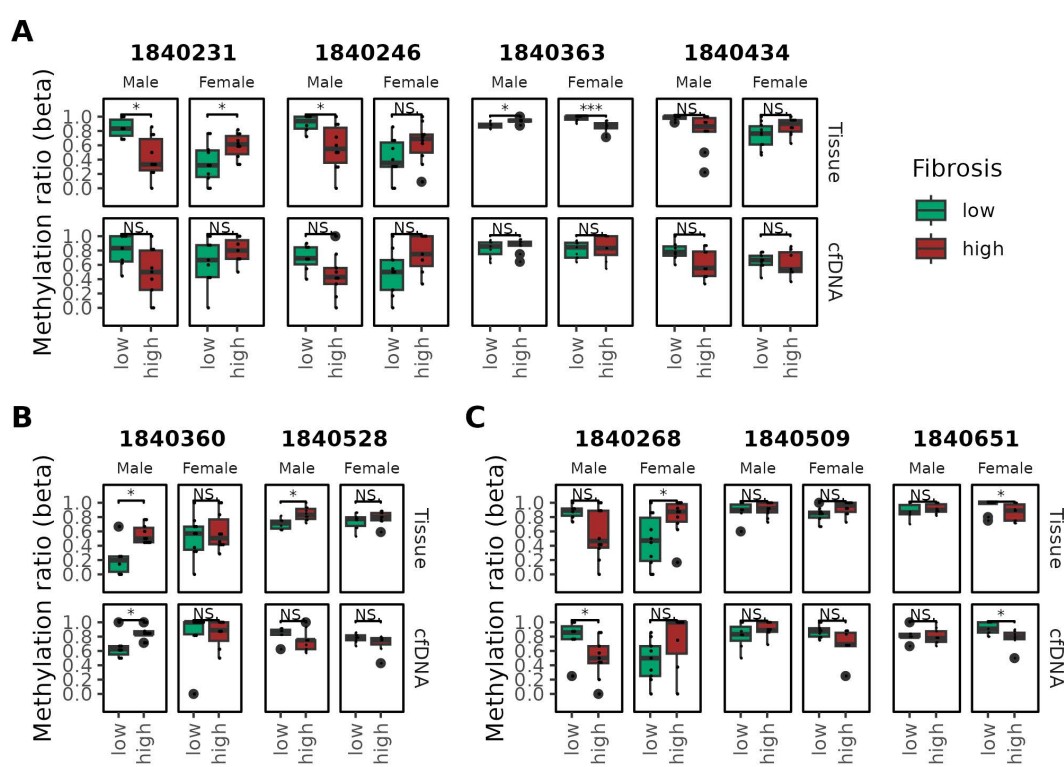

**Fig 3. Changes in methylation patterns according to fibrosis score in bins within the *PPARG* gene, stratified by group.** Fibrosis groups were classified as "low" for fibrosis scores less than 2 and "high" for scores of 2 or greater. P-values for comparisons between groups were obtained using the Wilcoxon rank-sum test. "*" and "***" indicate p-values less than 0.05 and 0.001, respectively. Cases with no statistically significant difference are labeled as "NS". Bin indices are labeled above each plot for clarity. (A) Methylation patterns in tissue and cfDNA for bins assigned to the "Tissue" group. (B) Methylation patterns for bins in the "Male" group. (C) Methylation patterns for bins in the "Female" group.

1840651) showed statistically significant differences in both tissue and cfDNA, displaying a pattern of hypomethylation with increasing fibrosis (Fig 3C). Taken together, these findings indicate that methylation patterns vary by both genomic location and sex.

## Methylation changes observed in cfDNA indicate an association with the Wnt signaling pathway

We performed pathway analysis on bins assigned to the "cfDNA" group, focusing on those located near TSS or within CpG islands, following the same approach used for the "Male" and "Female" groups. Although the results showed no clear enrichment trends, cross-validation with other databases consistently revealed associations with the Wnt signaling pathway (S5 Table).

## Intergenic methylation changes reflect alterations in the surrounding microenvironment of MASLD

We also conducted a comparison of CpG bins located in intergenic regions. Because the specific roles of individual intergenic CpG bins are difficult to determine, we used chromHMM status to compare the change proportion between overall chromatin states. Among bins not associated with fibrosis, those located in intergenic regions were used as a reference for comparison. Intergenic bins that *were* associated with fibrosis showed a reduced proportion of heterochromatin states in both GM12878 and HepG2 cells (Fig 4). In contrast, regions with increased proportions were primarily located in areas that transitioned between enhancer and heterochromatin states, or vice versa. While other chromatin state combinations with high enrichment ratios were also observed, they involved a limited number of bins.

Pathway analysis using the nearest genes to these highly enriched regions revealed a strong representation of hormone-related pathways, including those for progesterone, parathyroid hormone, estrogen, and androgen. Additionally, pathways related to T cell function (e.g., "IL-2/STAT5 signaling" [37]), liver-specific processes (e.g., "Fatty Acids and Lipoproteins Transport in Hepatocytes"), and potential fibrosis-associated pathways such as "Epithelial Mesenchymal Transition" [38] were also identified (S6 Table).

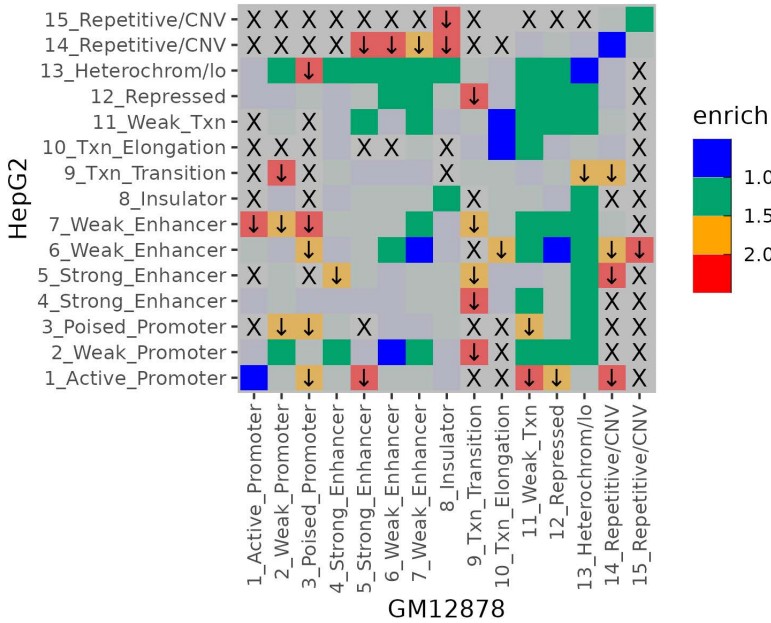

**Fig 4. Enrichment of chromatin state proportions in HepG2 relative to GM12878 for fibrosis-associated intergenic bins.** ChromHMM data for GM12878 and HepG2 were obtained from the ENCODE project via the UCSC Genome Browser database. Definitions for each chromatin state followed those described in the original reference publication [39]. An "X" indicates that no fibrosis-associated bins with the corresponding chromatin state transition were observed. Transitions involving fewer than 30 bins are shown with faded markings, while those enriched by more than 1.5-fold are marked with downward arrows.

## Cell type deconvolution analysis of tissue and cfDNA suggests an increase in T cell proportion in tissue and a corresponding increase in hepatocyte-derived cfDNA

In addition to pathway analysis, we conducted cell type deconvolution on MASLD samples to gain further insight into the underlying biological context (Fig 5). For comparison, reference data from healthy cfDNA and hepatocyte tissue were included, and available data from liver cirrhosis (LC) tissue and hepatocellular carcinoma (HCC) cfDNA and tissue were compared alongside MASLD samples. The comparison between MASLD and HCC tissues revealed marked differences in cell type composition, particularly in smooth muscle cells, kidney epithelium, pancreas alpha cells, liver hepatocytes, and skeletal muscle cells (S7 Table). In contrast, liver cirrhosis (LC) showed only a modest difference, limited to pancreas ductal cells. Comparison of cfDNA between MASLD and healthy controls revealed a highly significant difference in hepatocyte proportions (p < 0.001), while no statistically significant differences were observed for the other cell types.

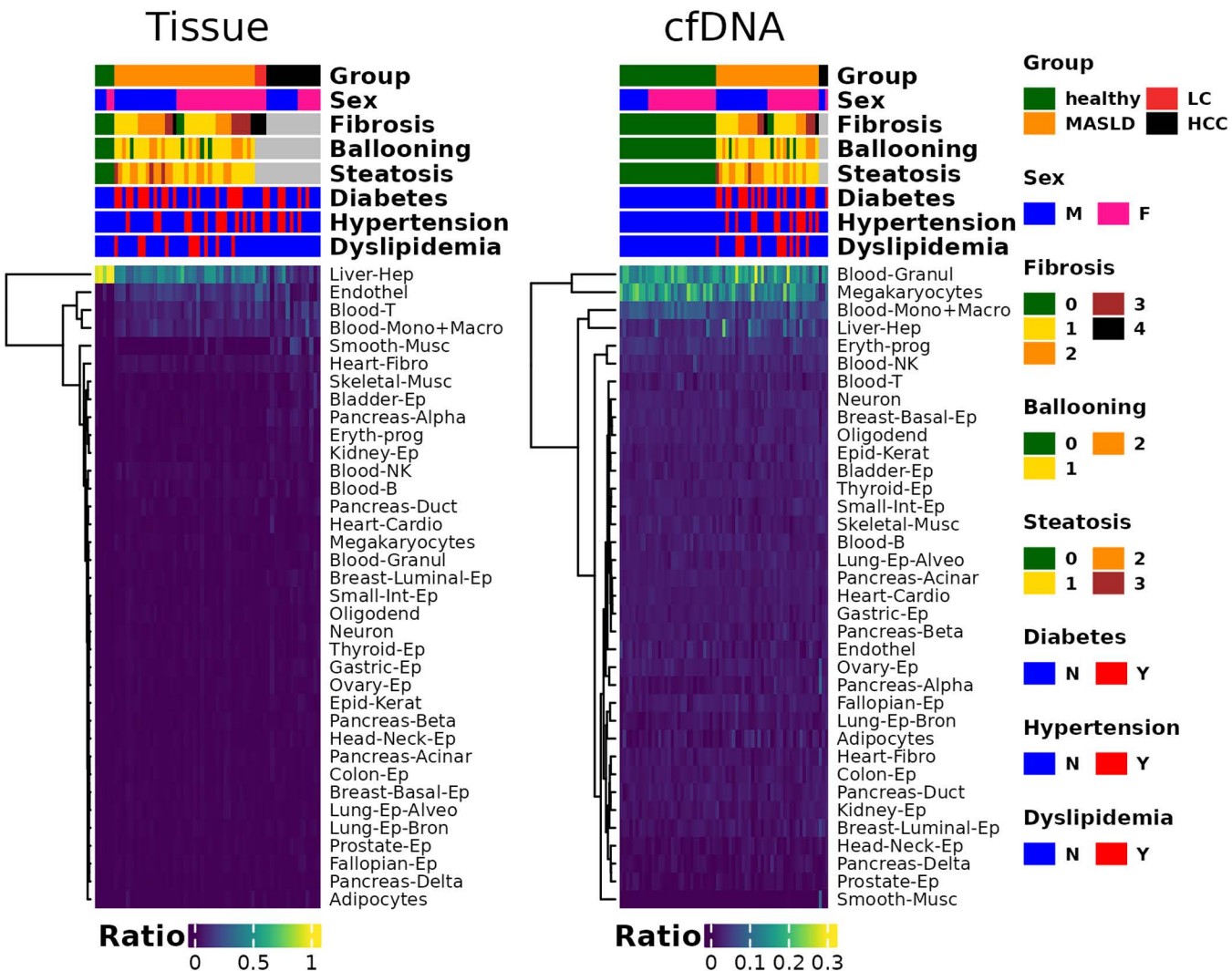

**Fig 5. Cell-type deconvolution results for tissue and cfDNA samples.** A heatmap was generated from the hierarchical clustering of cell-type deconvolution results. Abbreviations for each cell type are defined in the original publication from which the cell type deconvolution data were derived, as well as in S7 Table, which includes a copy of the cell type list used in that study.

Next, we performed multivariable regression analysis on MASLD samples to identify clinical variables associated with increased proportions of specific cell types. In tissue samples, ballooning was found to be strongly associated (p < 0.01) with elevated proportions of monocytes and T cells, while dyslipidemia and hypertension were closely linked to increased proportions of smooth muscle cells (S8 Table). In cfDNA samples, ballooning was strongly associated with an increased contribution from liver hepatocytes. Unexpectedly, fibrosis was also associated with differences in the proportion of cfDNA originating from bladder epithelial cells.

Among the cell types that showed significant associations and were separately clustered in Fig 5 via hierarchical clustering, we examined changes in cell type ratios according to fibrosis severity. In tissue samples, among T cells, monocytes/macrophages, and endothelial cells, only T cells exhibited a significant increase in proportion with higher fibrosis scores (Fig 6A). In cfDNA samples, of the two significantly associated cell types—granulocytes and hepatocytes—only hepatocytes showed a statistically significant increase (Fig 6B).

### cfDNA reflects known methylation pattern changes and holds potential for fibrosis classification

Finally, we then aimed to verify whether the methylation changes in the identified fibrosis-associated CpG sites aligned with previously known information at the individual gene level.

We focused on *TM4SF5* as a gene of interest because it is known to be expressed in hepatocytes in MASLD and has been previously reported to be associated with fibrosis [40], aligning well with our observed findings. Although determining the functional implications of methylation can be challenging, it has been well established that methylation around the CpG island at the TSS region and the first intron/exon often exhibits an inverse correlation with gene expression [41]. Therefore, we focused our exploration on these regions.

For TM4SF5, a reduction in methylation was observed in the cfDNA at a site located within the first intron (bin index: 9029246). When compared with the ENCODE data, the HepG2 liver cancer cell line showed hypomethylation at a nearby position within the same intron (cg11482108), in contrast to the GM12878 cells and hepatocytes (Fig 7A). Interestingly, two patients in the low fibrosis group who exhibited pronounced hypomethylation had a fibrosis grade of 0 but presented with steatosis, and both were female.

Furthermore, we investigated whether fibrosis groups could be distinguished based on methylation changes observed in cfDNA. Beta values from cfDNA corresponding to bins assigned to the "All" group were used as input features. A model trained on a subset of MASLD samples (excluding those reserved for validation) was tested on a mixed dataset comprising healthy cfDNA and the held-out MASLD samples. The model achieved an accuracy of approximately 0.85 (Fig 7B).

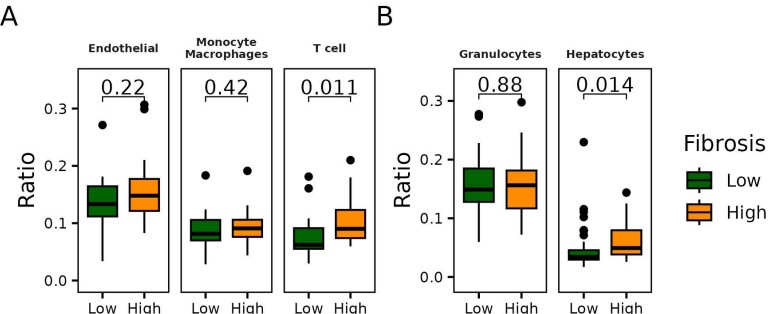

**Fig 6. Changes in the proportions of key cell types according to fibrosis progression. (A)** Distribution of cell types in tissue that showed significant differences in both hierarchical clustering and multivariable regression, stratified by fibrosis severity. Fibrosis was classified as "Low" for scores less than 2 and "High" for scores of 2 or greater. **(B)** Results of the analysis performed in cfDNA, mirroring the approach used in tissue. Using the same fibrosis group classification as in tissue.

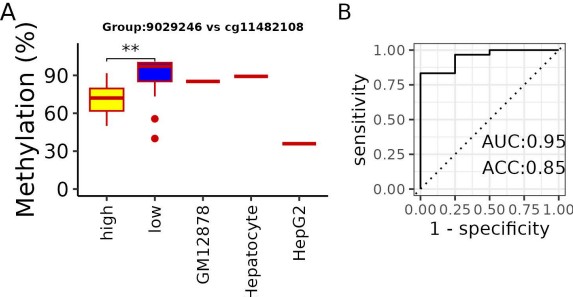

**Fig 7. Additional analyses based on cfDNA methylation data.** (A) Methylation pattern at the first intron of *TM4SF5*. Methylation patterns of control (GM12878), normal liver (hepatocyte), and liver cancer (HepG2) cells around fibrosis-associated CpG sites were compared. Methylation changes at the 450K site uniquely overlap with the first intron of the TM4SF5 gene. Fibrosis groups "low" and "high" were defined using the same criteria as described previously. *p*-value < 0.01 = ** (B) Classification results using methylation levels of CpG bins associated with fibrosis across all groups ("All"). AUC: area under the curve; ACC: accuracy.

## Discussion

In this study, we performed various group comparisons of WGMS patterns in liver biopsy tissues and blood samples from patients with MASLD according to fibrosis stages. The grouped fibrosis-associated methylation CpG clusters demonstrated different associations with MASLD depending on sex, cfDNA/tissue origin, and their genetic position. We observed an increase in liver-derived DNA in cfDNA, along with an increased proportion of T cells in liver tissue, aligning with previously known patterns of fibrosis in MASLD.

However, this study has several limitations. First and foremost, because liver biopsy is not recommended for most patients with MASLD [42], obtaining many samples was challenging. While we aimed to acquire pure healthy liver tissue WGMS data for use as a control, obtaining such data proved to be nearly impossible. Therefore, we had to use hepatocyte data from deconvolution studies [10] as substitutes. The hepatocyte data, obtained through fluorescence-activated cell sorting (FACS)-based separation, did not reflect the complexity of actual liver cells, and the healthy cfDNA samples not fully curated about fibrosis stages, meaning that they could not be considered fully healthy. Misclassifications observed in predictions from the trained classifier at Fig 7B may be attributable to these limitations. In addition, although age is known to have a substantial impact on DNA methylation [43], it was not explicitly accounted for in our analysis. This decision was primarily due to the limited sample size, which prevented us from obtaining statistically meaningful subgroup analyses. Moreover, age-related changes are often accompanied by complex physiological transitions—such as menopause—that cannot be fully captured by chronological age alone. Therefore, age was not included as a covariate in this study. Despite these experimental limitations, our results remain significant, as they align with previously identified pathways and trends related to TM4SF5 for MASH-associated fibrosis [44–46]. These results indicate the validity of our findings, and future studies with larger tissue samples may yield even clearer results.

Unlike traditional studies that focus on selecting a few precise CpG sites [47], we aimed to observe as many trends as possible, selecting significant sites through group comparisons (Fig 2). Although selecting only significant sites can help identify markers for testing, it limits the ability to observe broader changes, reducing the significance of conducting WGMS. Our sex-stratified pathway analysis (S4 Table) revealed a prominent enrichment of hormone-related pathways, consistent with previous observations suggesting that sex-specific differences may be linked to hormonal regulation. A key finding from the pathway analyses is that they support the cell type-specific patterns identified through deconvolution. Specifically, tissue samples showed an increased proportion of immune cells, while cfDNA samples reflected a higher contribution from hepatocytes (S8 Table). This is further corroborated by the enrichment of immune-regulatory pathways

such as the cGMP signaling pathway [48] (Table 2) and the Hippo pathway [49] (S3 Table) in tissue, and the detection of hepatocyte metabolism-related pathways such as Hedgehog and Wnt signaling in cfDNA (S5 Table), thereby strengthening the biological plausibility of our findings.

Our deconvolution results showed an increase in T-cell proportions with fibrosis progression in tissue, consistent with findings from previous studies [50]. However, we used reference tables from previous studies for cell type classification [10], which grouped various T-cell subtypes together and did not separately classify HSCs. Future studies could achieve more refined results by using FACS-sorted references for T-cell subtypes, Kupffer cells, and HSCs.

CpG hypomethylation in the first intron has previously been associated with increased gene expression [41]. Therefore, the observed hypomethylation by fibrosis at the TM4SF5 first intron CpG bin in cfDNA aligns with increased expression, shown in earlier reports on MASLD and murine and human fibrotic liver tissues [44,46]. ENCODE data further support this finding, as hypomethylation was observed in HepG2 liver cancer cells compared with normal hepatocytes and GM12878 at nearby first intron CpG sites.

We demonstrated that cfDNA methylation patterns can be used to classify fibrosis severity, utilizing bins assigned to the "All" group. This approach was chosen because consistency across samples was considered more critical than group-specific biological meaning for biomarker development. Despite the inherent limitations-such as a small sample size, group imbalance, and the use of a relatively simple model-we achieved a notably meaningful level of accuracy. These results suggest that future studies incorporating larger, more rigorously curated datasets may yield even more robust and reliable classification performance.

## Conclusion

Our WGMS analysis offers a comprehensive view of CpG methylation changes associated with MASLD and fibrosis progression. This study may lead to more non-invasive diagnosis of MASLD-associated fibrosis and their biomarker identification, which can be benefits for clinical purposes. We confirmed that various known MASLD-related changes significantly impact methylation patterns in not only tissue but also cfDNA. These findings thus suggest the potential to standardize the complexity of MASLD analyses, which have previously been conducted separately and lacked integration between tissue and blood samples, while also highlighting the possibility of non-invasive diagnosis through cfDNA.

## Supporting information

**S1 Table. Information on collected samples used in this study.**
(XLSX)

**S2 Table. Detailed information on the 87 selected bins.**
(XLSX)

**S3 Table. Pathway analysis results (KEGG) based on the gene list selected under additional filtering criteria.**
(XLSX)

**S4 Table. Pathway analysis results stratified by sex.**
(XLSX)

**S5 Table. Pathway analysis results based on cfDNA methylation data.**
(XLSX)

**S6 Table. Pathway analysis results based on intergenic bin methylation data.**
(XLSX)

**S7 Table. Statistical significance of cell type–specific proportion differences across disease types in tissue and cfDNA.**
(XLSX)

**S8 Table. Clinical variables identified as significantly affecting cell type proportions, along with the corresponding cell types.**
(XLSX)

## Author contributions

**Conceptualization:** Tae-You Kim.

**Data curation:** Jongseong Ahn, Hwang-Phill Kim.

**Formal analysis:** Jongseong Ahn, Soyeon Kim, Hwang-Phill Kim, Sang-Hyun Song.

**Funding acquisition:** Hwi Young Kim, Tae-You Kim, Jung Weon Lee.

**Investigation:** Jongseong Ahn.

**Methodology:** Jongseong Ahn, Soyeon Kim, Sunghoon Heo, Kyung-hee Pyo, Eun-Ae Shin, Wonsik Kim, Jae-Ho Lee, Hwang-Phill Kim, Sang-Hyun Song.

**Project administration:** Tae-You Kim, Jung Weon Lee.

**Resources:** Jae Yoon Jeong, Na Ryung Choi, Han Ah Lee, Hwi Young Kim.

**Software:** Sunghoon Heo.

**Supervision:** Hwi Young Kim.

**Visualization:** Hwi Young Kim, Jung Weon Lee.

**Writing – original draft:** Jongseong Ahn, Jung Weon Lee.

**Writing – review & editing:** Jung Weon Lee.

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
