## [Decision Letter · Decision Letter 0]

Dear Dr. Lee,

Thank you for submitting your manuscript to PLOS ONE. After careful consideration, we feel that it has merit but does not fully meet PLOS ONE’s publication criteria as it currently stands. Therefore, we invite you to submit a revised version of the manuscript that addresses the points raised during the review process.

We look forward to receiving your revised manuscript.

Kind regards,

Samuel O. Antwi, Ph.D.

Academic Editor

PLOS ONE

 [This work was supported by the National Research Foundation of Korea (NRF) funded by Basic Science Research Program through the National Research Foundation of Korea (NRF) funded by the Ministry of Science, ICT & Future Planning (NRF-2018R1A5A2025286 to HYK, NRF-2017M3A9A7050610 to T-YK, and NRF-2020R1A2C3008993 and NRF-2021M3A9D3024752 to JWL).]. 

[This work was supported by the National Research Foundation of Korea (NRF) funded by Basic Science Research Program through the National Research Foundation of Korea (NRF) funded by the Ministry of Science, ICT & Future Planning (NRF-2018R1A5A2025286 to HYK, NRF-2017M3A9A7050610 to T-YK, and NRF-2020R1A2C3008993 and NRF-2021M3A9D3024752 to JWL).]

[This work was supported by the National Research Foundation of Korea (NRF) funded by Basic Science Research Program through the National Research Foundation of Korea (NRF) funded by the Ministry of Science, ICT & Future Planning (NRF-2018R1A5A2025286 to HYK, NRF-2017M3A9A7050610 to T-YK, and NRF-2020R1A2C3008993 and NRF-2021M3A9D3024752 to JWL).]. 

4. In the online submission form, you indicated that [Raw data for this study were generated at IMBdx, Inc. Upon requests written to the corresponding author, the data and materials can be available.].

Reviewers' comments:

Reviewer's Responses to Questions

**Comments to the Author**

1. Is the manuscript technically sound, and do the data support the conclusions?

Reviewer #1: Partly

Reviewer #2: Partly

2. Has the statistical analysis been performed appropriately and rigorously?

Reviewer #1: Yes

Reviewer #2: Yes

3. Have the authors made all data underlying the findings in their manuscript fully available?

Reviewer #1: Yes

Reviewer #2: Yes

4. Is the manuscript presented in an intelligible fashion and written in standard English?

Reviewer #1: No

Reviewer #2: Yes

Reviewer #1: The present manuscript aimed at associating DNA methylation signatures, either at tissue or circulating level, with fibrosis in MASLD patients. Overall the effort of this study is clear. However, there are issues related to manuscript presentation and experimental design that need to be addressed.

-The selection of the cohort lacks important information on the characterization, including medium age and comorbiditues. As a matter of fact, especially when looking at systemic (i.e. blood) signatures, the occurrence of comorbidities could be relevant to properly interprete data (considering, for instance, the dissertation on neruodegenerative pathways or on T2D). Were the "non-MASLD" subjects employed male and female? Did the cohort of the employed GEO dataset mirror the features of the study cohort in terms of age, sex distribution?

-Considering the high incidence of MASLD in worldwide population, was a a priori power analysis performed in order to predict if the considered sample size in the present study was appropriate to reach significant reliable results?

-Cf-DNA is generally fragmented. Does this issue affects the ability to perform a proper cell typedeconvolution analysis encompassing cfDNA data?

-I suggest authors to organize the Results section subdividing into results related to tissue and to cfDNA for the sake of clarity

-The Figure captions should be moved since they make difficult to read the results section

-In the Discussion, a brief illustration of the role of age (according to data related to the study cohort) should be provided, since it DNA methylation notably is modulated during aging. Therefore, it is possible that certain signatures may be related to MASLD patients with a specific range of age.

-Eventually, authors could perform some metric performance analysis (such as evaluation of accuracy) to see if specific methylation signatures could represent candidate biomarkers.

-The manuscript should be revised for some typos (for insatnce, "abstrct").

Reviewer #2: The authors' starting point was "Analysis of cfDNA through DNAm patterns can reveal tissue-specific DNAm profiles...lines 91-92". This is potentially a very interesting starting point and the authors analysed tissue and blood samples from patients and controls (blood only in controls) to address this point which is quite novel. However the manuscript is written in such a complicated manner that it is not clear what has been done with these samples and what actually has been found. The really interesting data from patients/controls are submerged under secondary data or data analysis from databases, cell lines and other irrelevant resources. I would like to encourage the authors to consider rewriting the whole manuscript with focus on their own data. Then they can use secondary data to test the hypotheses they create from their primary data. I can see that they identified some possible biomarkers (which was indeed the whole point of the study) such as PAX3 and FASN but these are again buried under over-analysis and not even mentioned in the abstract. The conclusions reached and mentioned in the abstract should be specific, not vague.

**Do you want your identity to be public for this peer review?** For information about this choice, including consent withdrawal, please see our Privacy Policy

Reviewer #1: No

Reviewer #2: No

---

## [Author Response · Author response to Decision Letter 1]

29 May 2025

Author responses to reviewers’ comments

Reviewer #1: The present manuscript aimed at associating DNA methylation signatures, either at tissue or circulating level, with fibrosis in MASLD patients. Overall the effort of this study is clear. However, there are issues related to manuscript presentation and experimental design that need to be addressed.

Response: Thank you for your comment. In response, we have made comprehensive revisions to the manuscript in line with your suggestions. The text has been refined to be more concise and logically structured, and repetitive analytical content that was deemed excessive has been removed. We believe these changes have improved the overall clarity and flow of the manuscript.

Comment #1: The selection of the cohort lacks important information on the characterization, including medium age and comorbiditues. As a matter of fact, especially when looking at systemic (i.e. blood) signatures, the occurrence of comorbidities could be relevant to properly interprete data (considering, for instance, the dissertation on neruodegenerative pathways or on T2D). Were the "non-MASLD" subjects employed male and female? Did the cohort of the employed GEO dataset mirror the features of the study cohort in terms of age, sex distribution?

Response #1: Thank you for your comment. In response, we have incorporated additional characterization data that may influence MASLD, as detailed in Figure 1 and S1 Table. The GEO dataset, which consists of purified hepatocytes, was used solely as a positive control for cell type deconvolution analysis and was not included in the direct comparisons.

Following your suggestion, we excluded the ambiguous non-MASLD group from the analysis and instead included liver cirrhosis (LC) and hepatocellular carcinoma (HCC) groups for comparative purposes. However, due to limited clinical information, these groups were used exclusively in the cell type deconvolution analysis. Further details have been provided in the Methods section and S1 Table.

Comment #2: Considering the high incidence of MASLD in worldwide population, was a a priori power analysis performed in order to predict if the considered sample size in the present study was appropriate to reach significant reliable results?

Response #2: Thank you for your comment. We would like to clarify that our study is not a prospective cohort study; rather, MASLD cases were collected after diagnosis. Therefore, considerations regarding incidence rates may be somewhat outside the scope of this work.

Additionally, due to the epigenetic nature of methylation, the patterns observed tend to vary across individuals rather than following a fixed distribution. These variations differ by genomic locus, making it difficult to estimate in advance the required sample size based on expected changes at unspecified CpG sites. Nonetheless, based on forced approximations using results from a previous study investigating the association between methylation and fibrosis (https://doi.org/10.1186/s13148-021-01129-y), we estimate that approximately 800 samples would be required—an impractically large number for the present study.

As discussed in the manuscript, we acknowledge the need for larger sample sizes to strengthen future analyses. However, we believe the findings from our current dataset still offer meaningful insights and contribute valuable preliminary evidence to the field.

Comment #3: cfDNA is generally fragmented. Does this issue affects the ability to perform a proper cell type deconvolution analysis encompassing cfDNA data?

Response #3: Thank you for your comment. Fundamentally, except for long-read sequencing platforms, current sequencing technologies detect DNA primarily in the form of fragments that are comparable in size to, or only slightly longer than, those of cfDNA. Accordingly, most deconvolution methods have been designed from the outset to accommodate fragmented DNA.

While this does not guarantee equal detection efficiency across all cell type markers or target regions, such limitations are inherent to the nature of methylation-based analysis itself rather than a specific consequence of using cfDNA. Therefore, the use of cfDNA fragments does not introduce an additional bias beyond this intrinsic constraint.

Comment #4: I suggest authors to organize the Results section subdividing into results related to tissue and to cfDNA for the sake of clarity

Response #4: Thank you for your suggestion. The manuscript has now been reorganized in the following order: tissue-level analysis, followed by sex-specific analyses (in both tissue and cfDNA), and concluding with cfDNA-based findings.

Comment #5: The Figure captions should be moved since they make difficult to read the results section

Response #5: Thank you for your comment. However, according to PLOS ONE’s formatting guidelines, “figure captions should appear in the manuscript text in read order, immediately following the paragraph where the figure is first cited. Do not include captions as part of the figure files or submit them in a separate document.” Based on this policy, we have placed the figure captions accordingly within the manuscript. We anticipate that any remaining issues related to readability will be addressed during the editorial processing stage.

Comment #6: In the Discussion, a brief illustration of the role of age (according to data related to the study cohort) should be provided, since it DNA methylation notably is modulated during aging. Therefore, it is possible that certain signatures may be related to MASLD patients with a specific range of age.

Response #6: Thank you for your comment. In response, we have added the following explanation to lines 394–400 of the manuscript:

“In addition, although age is known to have a substantial impact on DNA methylation [43], it was not explicitly accounted for in our analysis. This decision was primarily due to the limited sample size, which prevented us from obtaining statistically meaningful subgroup analyses. Moreover, age-related changes are often accompanied by complex physiological transitions—such as menopause—that cannot be fully captured by chronological age alone. Therefore, age was not included as a covariate in this study.”

Comment #7: Eventually, authors could perform some metric performance analysis (such as evaluation of accuracy) to see if specific methylation signatures could represent candidate biomarkers.

Response #7: Thank you for your comment. The requested analysis has been incorporated and is now presented in Figure 7B.

Comment #8: The manuscript should be revised for some typos (for insatnce, "abstrct").

Response #8: Thank you for your comment. We have made comprehensive revisions to the manuscript, during which the relevant content has been updated accordingly.

Reviewer #2: The authors' starting point was "Analysis of cfDNA through DNAm patterns can reveal tissue-specific DNAm profiles...lines 91-92". This is potentially a very interesting starting point and the authors analysed tissue and blood samples from patients and controls (blood only in controls) to address this point which is quite novel. However the manuscript is written in such a complicated manner that it is not clear what has been done with these samples and what actually has been found. The really interesting data from patients/controls are submerged under secondary data or data analysis from databases, cell lines and other irrelevant resources. I would like to encourage the authors to consider rewriting the whole manuscript with focus on their own data. Then they can use secondary data to test the hypotheses they create from their primary data. I can see that they identified some possible biomarkers (which was indeed the whole point of the study) such as PAX3 and FASN but these are again buried under over-analysis and not even mentioned in the abstract. The conclusions reached and mentioned in the abstract should be specific, not vague.

Response: Thank you for your thoughtful and constructive feedback. In agreement with your comments, we undertook a comprehensive reanalysis and have revised the manuscript extensively. Excessive use of external data that risked diluting the main focus has been removed. The overall structure has been reorganized to follow a clearer narrative: tissue-level analyses, followed by sex-specific findings, and then cfDNA-based observations.

In the Discussion section, content that was overly detailed and potentially hindered clarity has been removed to improve readability. The manuscript now places greater emphasis on our own dataset, including the detailed examination of methylation patterns in specific genes such as PPARG and TM4SF5, as well as the addition of classification results based on cfDNA methylation. We believe these changes substantially improve the clarity and scientific relevance of the study.

---

## [Decision Letter · Decision Letter 1]

Whole genome DNA methylation patterns in tissue and cfDNA associated with fibrosis reflect the complex signature of MASLD

PONE-D-25-01339R1

Dear Dr. Lee,

We’re pleased to inform you that your manuscript has been judged scientifically suitable for publication and will be formally accepted for publication once it meets all outstanding technical requirements.

Kind regards,

Samuel O. Antwi, Ph.D.

Academic Editor

PLOS ONE

Additional Editor Comments (optional):

Reviewers' comments:

Reviewer's Responses to Questions

**Comments to the Author**

Reviewer #1: All comments have been addressed

Reviewer #2: All comments have been addressed

2. Is the manuscript technically sound, and do the data support the conclusions?

Reviewer #1: Yes

Reviewer #2: Yes

3. Has the statistical analysis been performed appropriately and rigorously?

Reviewer #1: Yes

Reviewer #2: Yes

4. Have the authors made all data underlying the findings in their manuscript fully available?

Reviewer #1: Yes

Reviewer #2: Yes

5. Is the manuscript presented in an intelligible fashion and written in standard English?

Reviewer #1: Yes

Reviewer #2: Yes

Reviewer #1: (No Response)

Reviewer #2: The manuscript has improved substantially. The authors addressed all my comments. No further comments.

**Do you want your identity to be public for this peer review?** For information about this choice, including consent withdrawal, please see our Privacy Policy

Reviewer #1: No

Reviewer #2: No

---

## [Editor Report · Acceptance letter]

PONE-D-25-01339R1

PLOS ONE

Dear Dr. Lee,

I'm pleased to inform you that your manuscript has been deemed suitable for publication in PLOS ONE. Congratulations! Your manuscript is now being handed over to our production team.

Kind regards,

on behalf of

Dr. Samuel O. Antwi

Academic Editor

PLOS ONE